# Exploring the Methane to Methanol Oxidation over Iron and Copper Sites in Metal–Organic Frameworks

**Francesco Tavani** † and **Alessandro Tofoni** † and **Paola D'Angelo** *

Department of Chemistry, Sapienza University, P.le A. Moro 5, 00185 Rome, Italy;
francesco.tavani@uniroma1.it (F.T.); alessandro.tofoni@uniroma1.it (A.T.)
* Correspondence: p.dangelo@uniroma1.it
† These authors contributed equally to this work.

**Abstract:** The direct oxidation of methane to methanol (MTM) is a significant challenge in catalysis and holds profound economic implications for the modern chemical industry. Bioinspired metal–organic frameworks (MOFs) with active iron and copper sites have emerged as innovative catalytic platforms capable of facilitating MTM conversion under mild conditions. This review discusses the current state of the art in applying MOFs with iron and copper catalytic centers to effectuate the MTM reaction, with a focus on the diverse spectroscopic techniques employed to uncover the electronic and structural properties of MOF catalysts at a microscopic level. We explore the synthetic strategies employed to incorporate iron and copper sites into various MOF topologies and explore the efficiency and selectivity of the MOFs embedded with iron and copper in acting as catalysts, as well as the ensuing MTM reaction mechanisms based on spectroscopic characterizations supported by theory. In particular, we show how integrating complementary spectroscopic tools that probe varying regions of the electromagnetic spectrum can be exceptionally conducive to achieving a comprehensive understanding of the crucial reaction pathways and intermediates. Finally, we provide a critical perspective on future directions to advance the use of MOFs to accomplish the MTM reaction.

**Keywords:** methane; methanol; MOF; spectroscopy; XAS; IR; Raman; Mössbauer; EPR; XPS

## 1. Introduction

Methane ($CH_4$) is a highly abundant hydrocarbon, being the main constituent of shale and natural gas (where it amounts to up to 80–90% of all available reservoirs), as well as biogas [1], a mixture of gases (mostly $CH_4$ and $CO_2$) produced from biodegradable materials, in an effort to implement alternative approaches for natural gas decarbonization [2]. Methane is also found in crystalline hydrates at the continental slopes of many oceans, as well as in permafrost areas, and may be catalytically converted to numerous industrially relevant compounds, such as methanol, synthesis gas, hydrogen cyanide, ethylene, formaldehyde, methyl chloride, methyl bromide and aromatics [3]. For these reasons, methane plays a key role in the modern chemical industry as a widespread and renewable energy and chemical source. However, methane is also the second strongest anthropogenic greenhouse gas after $CO_2$; it possesses a global warming potential (GWP) of about 28–36 over 100 years (the GWP of $CO_2$ is defined as equal to unity) [4], and its atmospheric burden has increased by more than twice since 1850 [1]. Methane emission sources include natural ones (321 Tg CH4 $yr^{-1}$), such as water, geological, insect, and animal sources, as well as anthropogenic sources (696 Tg CH4 $yr^{-1}$), which are mainly related to agriculture, petroleum, ruminants, and biomass and biofuel burning [5,6]. Moreover, methane emissions caused by leakages or incomplete flaring during fossil fuel production and usage are especially troublesome, while approximately 144 billion cubic meters of methane gas are annually burned causing severe consequences to the global carbon footprint [7]. As a consequence, innovative methodologies and materials are urgently needed to more efficiently convert

methane into useful energy or chemical sources and to mitigate global warming effects by decreasing methane emissions. Unfortunately, most of the current methane conversion processes suffer from the intrinsic inertness of the $CH_4$ molecule [8,9] and require relatively elevated temperatures and pressures [6,10].

In particular, the oxidation of methane to methanol (MTM) is a central reaction for the modern chemical industry, with methanol and its derivatives being obtained almost entirely from natural gas (∼65%) and coal (∼35%) [11]. Global demand for methanol has nearly doubled in the last decade, reaching 106 million tonnes in 2021, and is expected to continue to grow [4,12]. Current demand is based on the use of methanol as a chemical feedstock (>60%), predominantly to manufacture olefins (32%), formaldehyde (23%) and acetic acid (8%) [4]. In addition, transport fuels also employ methanol, such as through methyl tert-butyl ether and biodiesel [12]. Common MTM conversion routes—at an industrial scale—involve the initial transformation of methane from natural gas into syngas (a mixture of CO and $H_2$). Subsequently, syngas is converted to methanol by a heterogeneous catalytic reaction, typically conducted over a bimetallic (Cu/Zn or Cu/Cr) catalyst at 220–250 °C and high pressure (70–100 bar) [13,14]. The high temperature (>800 °C) and energy demands required by methane steam reforming are associated with elevated costs, and significant efforts have been devoted to developing alternative catalytic platforms to efficiently and sustainably accomplish the MTM oxidation reaction. Furthermore, it is not yet possible to directly convert methane to methanol through an industrial route [9,15]. The lack of catalysts for such a process is due to the relatively large energy barriers associated with the activation of the nonpolar and highly symmetric $CH_4$ molecule and the higher relative reactivity of the methanol product. In fact, methane possesses a negligible electron affinity, a high ionization energy, a very large HOMO-LUMO gap, and an elevated pKa value; it does not display a dipole moment or significant polarizability [9], and large energies are required for both homo- and heterolytic methane C–H bond cleavages. In addition, the C-H bonds of methanol have a 0.4 eV lower bond dissociation energy (BDE) than those in methane, making it challenging to prevent product overoxidation [15]. To date, a wide variety of solid-state systems have been employed to investigate conversion reactions of methane to methanol and other oxygenated products. These include molybdenum oxide-based catalysts, such as silica-supported $MoO_3$ [16], $MoO_x$ materials [17,18], $MoO_x$/La-Co-O [19], $VO_x$/$SiO_2$ [20], and microporous materials, such as Fe/ZSM-5 [21–24] and Cu-MOR [25–29] zeolites, as well as Au-based photocatalysts supported on $TiO_2$ [30,31]. However, one of the main issues of such heterogeneous catalysts is their reduced level of tunability, which severely limits how one may vary the chemical nature of the platform components while exploring the "chemical space" of effective MTM catalysts. Another issue affecting systems such as Fe- and Cu-exchanged zeolites is the heterogeneous metal speciation within the zeolitic framework, which is influenced by factors such as the original synthetic method, the employed activation procedure, and the total metal content [32,33]. For instance, existing studies on the Fe–ZSM-5 zeolite report a wide range of first-shell Fe–Fe, Fe–Fe, and Fe–Si/Al bond distances, due to the high heterogeneity in the metal site distribution that severely undermines efforts to decipher the properties of the true active sites [34].

While electro- [35] and photoelectrocatalytic [36,37] systems have also been proposed for methane oxidation, an emerging approach to tackle the MTM reaction, a "holy grail" in the field, makes use of metal–organic frameworks (MOFs). As detailed in the following sections, MOFs are unique porous materials with a very high potential for methane chemistry [6].

## 2. Metal–Organic Frameworks as Catalytic Platforms for the MTM Reaction

MOFs are a class of crystalline and permanently porous materials, which have been attracting significant attention in the field of heterogeneous catalysis [38]. MOFs are composed of inorganic metal clusters (referred to as secondary building units or SBUs) bound to polydentate organic ligands (referred to as linkers), resulting in highly ordered structures of well-defined topologies [39]. Figure 1 presents several of the most common

linkers, SBUs, and MOF structures, including some explored to accomplish the MTM reaction and discussed in this work. Owing to their hybrid nature, MOFs have sparked great interest for their catalytic applications. MOFs in fact possess an exceptional degree of structural and chemical tunability, significantly higher than those of other porous materials, such as zeolites. This property allows one to design MOFs that are specially tailored for a given application by finely tuning the MOF composition [6,38–46]. For instance, it was shown that the pore size of the UiO-66 MOF could be regulated by increasing the linker length, yielding the UiO-67 and UiO-68 structures [47]. Furthermore, a density functional theory (DFT)-based study demonstrated that the photocatalytic properties of MIL-125, a MOF containing Ti nodes and 1,4-benzene-dicarboxylate (BDC) moieties, were influenced by functional groups introduced on the terephthalate ligand, evidencing how the Ti band gap was reduced upon insertion of an amino group on the linker [48]. In addition, multivariate mixed-metal MOFs can be prepared to either exploit the interaction of adjacent metal sites improving catalytic activity [49], or to dilute the active metal site with the aim of increasing product selectivity [50]. MOF reactivity may also be tuned through a large number of post-synthetic modifications, which range from ligand or metal exchange to ligand functionalization strategies [51–54]. As an example, a bifunctionalized MIL-101(Cr)-$SO_3H$-$NH_2$ catalyst was prepared by a series of post-synthetic modification steps, and the MOF showed excellent yields for the one-pot deacetalization–nitroaldol reaction of benzaldehyde dimethyl acetal into (2-nitrovinyl)benzene with no loss of reactivity after three consecutive cycles [55]. MOFs can also be employed as catalytic platforms to support active single-atom metal sites or metal clusters [56–61]. For instance, sintering-resistant single Ni sites have been embedded into the Zr-based NU-1000 framework, acting as efficient ethylene hydrogenation catalysts [62]. Apart from exhibiting good thermal and chemical stability, MOF metal sites possess well-defined positions, local coordination geometries, oxidation, and spin states. This allows one not only to exert greater synthetic control over the MOF metal centers but also to more easily rationalize the catalytic activity of the given MOF material.

The first application of MOF catalysts for alkane C–H bond activation was reported almost a decade ago by Prof. Jeffrey Long et al. [50]. In this pioneering work, it was shown that MOF-74(Fe) catalyzes the $N_2O$-based direct conversion of ethane to ethanol. The reaction mechanism was later demonstrated to involve a Fe(IV)=O intermediate generated via the oxidation of Fe(II) sites by $N_2O$, producing ethanol through a C-H bond cleavage-radical rebound process [63]. Such novel findings motivated further studies to investigate the MOF-based oxidation of light alkanes, with the aim of achieving the grand challenge of converting methane to methanol over MOFs [64,65]. Indeed, the MTM reaction has been successfully accomplished both over iron and copper centers present in several different MOFs, as a result of attempts to mimic the catalytic activity of methane monooxygenase enzymes, which effectively perform the MTM reaction over iron and copper active sites. As far as Fe-based MOFs are concerned, MIL-100(Fe) and PCN-250 were proven to hydroxylate methane to methanol using $N_2O$ as the oxidant [34,66], while the mixed-metal MIL-53(Al, Fe) material was shown to carry out the $H_2O_2$-based MTM reaction [67]. In addition, the UiO-67 MOF has been employed as a catalytic platform to build a complex photocatalytic system based on a mono-iron hydroxyl site, able to perform the MTM conversion with 100% selectivity by activating $O_2$ in a liquid-phase process [68]. Active copper sites obtained through the insertion of copper dimers or clusters in highly stable Zr-MOFs also exhibit a promising degree of methane hydroxylation activity. For instance, MOF-808 has been employed as a platform containing bioinspired active copper dimeric sites that react to $CH_4$ and $N_2O$ to yield methanol [69]. Cu-oxo dimers and clusters have also been embedded in the Nu-1000 framework through water-mediated cation exchange and atomic layer deposition techniques [70,71], allowing the activation of $O_2$ for the MTM reaction in mild conditions.

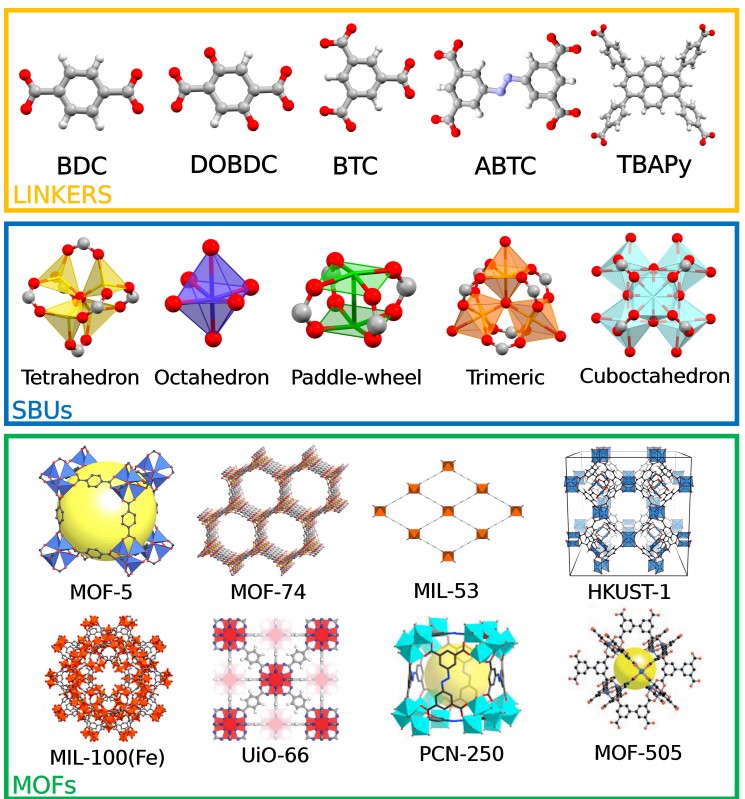

**Figure 1.** Top: common organic MOF linkers. From left to right: benzene-1,4-dicarboxylate (BDC), 2,5-2,5-dioxido-1,4-benzenedicarboxylate (DOBDC), benzene-1,3,5-tricarboxylate (BTC), 3,3',5,5'-azobenzene-tetracarboxylate (ABTC), 1,3,6,8-tetrakis(p-benzoate)-pyrene (TBAPy). Color code: oxygen, red; carbon, grey; nitrogen, blue; hydrogen, white. Middle: common MOF secondary building units (SBUs). Bottom: common MOF structures. MOF-5 (Reprinted with permission from Kaye et al., J. Am. Chem. Soc. 2007, 129, 46, 14176–14177 [72]. Copyright 2007 American Chemical Society), MOF-74 (Reprinted with permission from Bloch et al., J. Am. Chem. Soc. 2011, 133, 37, 14814–14822 [73]. Copyright 2011 American Chemical Society), MIL-53 (reproduced from Bara et al., Mater. Horiz. 2021, 8, 12, 3377–3386 [74]. Copyright 2021 Royal Society of Chemistry), HKUST-1 (reproduced from Hendon and Walsh, Chem. Sci. 2015, 6, 7, 3674–3683 [75]. Copyright 2015 Royal Society of Chemistry), MIL-100(Fe) (reproduced with permission from Quijia et al., J. Drug. Deliv. Sci. Technol. 2021, 61, 102217 [76]. Copyright 2021 Elsevier), UiO-66 (Reprinted with permission from Cavka et al., J. Am. Chem. Soc. 2008, 130, 42, 13850–13851 [47]. Copyright 2008 American Chemical Society), PCN-250 (reproduced with permission from Yuan et al., Joule 2017, 1, 4, 806–815 [77]. Copyright 2017 Elsevier), MOF-505 (Adapted with permission from Chen 2005, figure 1a, p. 4745 [78]. Copyright 2005 Oxford University Press).

In the following sections, we discuss the studies that have investigated the use of iron and copper centers anchored in MOFs for the MTM conversion, as well as the main spectroscopic techniques employed to unravel the properties of the key reaction active sites and intermediates.

## 3. MOFs Exhibiting Iron Active Sites for the MTM Conversion

Iron is a low-cost metal with reduced toxicity that can shuttle between different oxidation states. Furthermore, iron may form both mononuclear and polynuclear biomimetic clusters that, when integrated into MOFs, show high promise for the oxidation of methane. The application of Fe-based MOFs as catalysts for methane oxidation has been inspired by the presence of iron in catalytic enzyme pockets, such as the soluble methane monooxygenase (sMMO) active site. Methanotrophic bacteria employ methane as their sole energy source and utilize sMMO to oxidize methane to methanol at the initial phase of methane

metabolism [4]. In particular, the sMMO enzyme is constituted by (i) a hydroxylase (MMOH), which is responsible for MTM conversion, (ii) a reductase (MMOR) that guides the activated $O_2$ towards the hydroxylase core, and (iii) a regulatory protein (MMOB) controlling $CH_4$ admission to the MMOH active pocket. Notably, the sMMO active site possesses a hydrophobic cavity running through the protein center, which favors the release of methanol, being hydrophilic, and prevents its overoxidation. The active site for the oxidation of methane, solved recently by Banerjee and coworkers [79], is known as compound Q and is built from a diiron cluster, as shown in Figure 2. At the beginning of the sMMO MTM catalytic cycle, two electrons are transferred in the presence of a nicotinamide adenine dinucleotide cofactor (NADH) to the $H^{ox}$ intermediate, leading to the reduction of the Fe(III)-Fe(III) antiferromagnetically coupled, high spin, dimeric site to a Fe(II)-Fe(II) species ($H^{red}$ intermediate) [80], as shown in Figure 3.

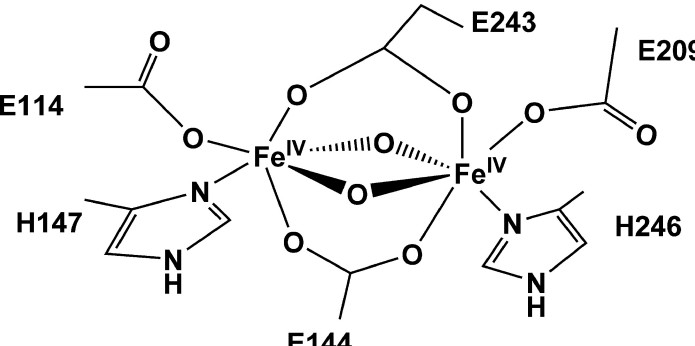

**Figure 2.** Proposed Fe(IV) diiron structure of compound Q in sMMO. The numbers denote amino acids in the side chains: H, histidine; E, glutamate (reprinted with permission from Ref. [81]. Copyright 2017, American Chemical Society).

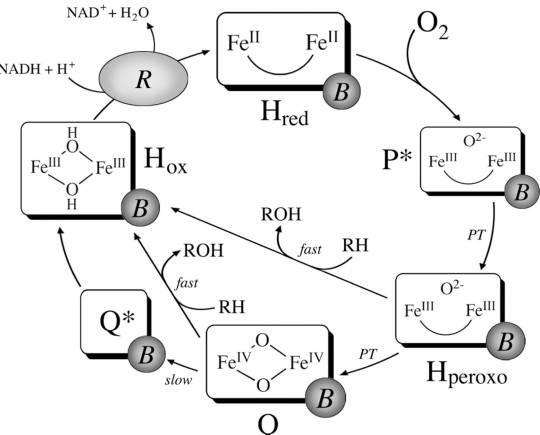

**Figure 3.** Schematic depiction of the sMMO catalytic cycle. R represents the MMOR reductase component, while B is the regulatory MMOB component (reprinted with permission from Ref. [80]. Copyright 2015, American Chemical Society).

Subsequently, the MMOH-MMOB complex guides $O_2$ to the diiron active site, leading to the formation of intermediate P* [82,83]. Molecular oxygen is then activated and the O–O bond is cleaved to yield compound Q, which is responsible for the hydroxylation of methane. Specifically, it has been proposed that the H-atom abstraction from $CH_4$ proceeds via the formation of a hydroxyl radical, bound to a partially reduced Fe(III)-Fe(IV) cluster, and a methyl radical, with the two radicals recombining in the methanol formation step [84]. Notably, sMMO possesses unrivaled selectivity by enclosing the hydrophobic $CH_4$ molecule in a nonpolar environment near the active site that promotes the radical rebound mechanism and facilitates product removal. Furthermore, the magnetic properties of the sMMO Fe(III)-Fe(III) dimer are believed to be crucial for the enzymatic

reactivity [85]. It is important to note that while performing the MTM conversion, all sMMO compartments display high surface area and porosity, while leveraging an Fe-based active site with interchanging oxidation states in a concerted and hierarchical fashion. These are all properties that can be mimicked using MOFs, and several studies have been carried out to explore the MTM conversion over this highly tunable class of artificial materials.

In a work published in 2018, Gascon and coworkers [86] utilized isolated Fe sites incorporated into the MIL-53(Al,Fe) MOF framework (Figure 4A) to carry out the MTM conversion in the presence of $H_2O_2$. The MIL-53(Al) structure is built by chains of $AlO_6$ octahedra connected by bridging BDC linkers and trans-located $HO^-$ ions [87]. This MOF was chosen as the support matrix due to the inertness of the $AlO_6$ units towards redox reactions and since its internal pore cavities are built from a hydrophobic organic linker that could favor rapid methanol desorption. Two distinct synthetic strategies were pursued to incorporate Fe sites into the Al-based MOF framework, namely, a post-synthetic cation exchange by placing into contact MIL-53(Al) and different $FeCl_3$ solutions (HTS samples), as well as an electrochemical synthesis of MIL-53(Al) from an Al electrode and a terephthalic acid solution containing $FeCl_3$ aliquots (ECS samples) [86]. It was found that between 0.15 and 2 wt % (wt %), Fe could be successfully incorporated into MIL-53(Al) using cation exchange at a relatively low temperature (80 °C), while electrochemical synthesis allows between 0.3 and 5.5 wt % Fe incorporation. In these samples with low Fe-loading (LL samples), small MOF nanoparticles form by agglomeration and deplete the well-known breathing effect of the MIL-53 material [87]. Catalytic methane oxidation tests were performed in water using $H_2O_2$ as the oxidant over both the HTS and ECS samples at temperatures below 60 °C, leading—in both cases—to selective methanol formation with no C–C coupling products [86]. Conversely, performing the post-synthetic Fe incorporation at a higher temperature (120 °C) allowed the introduction of larger Fe amounts (up to 16.6 wt %) into high Fe-loading samples (HL samples) but also yielded extraframework $Fe_2O_3$, and led to the formation of a higher amount of side products during the methane oxidation catalytic tests. Altogether, the ECS samples were the most catalytically active, with selectivities towards oxygenates of about 80% (see Figure 4B), while suppressing the formation of undesired extraframework iron species, e.g., $Fe_2O_3$. [1]H-NMR and gas chromatography analyses evidenced the formation of methanol, methyl peroxide, formic acid, and $CO_2$ as sole reaction products [86]. As detailed in Section 5, a series of different spectroscopic techniques were employed to better understand the nature and stability of the iron catalytic sites.

In 2020, Bollini and coworkers [34] reported the successful low-temperature MTM oxidation over MIL-100(Fe), a MOF comprised of tri-iron SBUs first discovered by Gérard Férey et al. [88,89]. In particular, MIL-100(Fe) is made up of trimeric iron nodes interconnected by trimesate linkers to produce a framework of overall MTN topology, as shown in Figure 5A. In the MOF SBU, there are three Fe sites bridged by a $\mu$-oxo center and coordinated in an octahedral environment, with water molecules located in two-thirds of the apical coordination positions, while a monovalent anion (such as $HO^-$ or $F^-$) occupies the third available site. Notably, thermally treating hydroxyl-containing MIL-100(Fe) in vacuum or under an inert gas flow generates open Fe(III) or Fe(II) sites via the elimination of water molecules or of $HO^-$ groups, respectively [34,89]. By co-feeding $CH_4$ and the $N_2O$ oxidant at 200 °C over a Fe(II)-containing activated MOF sample, and then exposing the post-reaction sample to a He stream containing water, the production of 0.34 mol (mol Fe)$^{-1}$ of methanol was observed. Interestingly, this value was quite similar to the cumulative moles of reacted methane (0.32 mol (mol Fe)$^{-1}$) [34]. The produced quantity of methanol was also close to a value of 1 mole of $CH_3OH$ per mole of Fe(II), whose abundance in the MOF was found to approach the theoretical maximum density. Furthermore, by employing NO as a selective probe of the Fe(II) sites, it was demonstrated that methanol formation occurs exclusively over Fe(II) and not Fe(III) centers. Finally, the formation of methoxy intermediates within MIL-100(Fe), which react with water to yield gas-phase methanol,

was strongly supported by exposing the post-reaction MOF to increasing fractions of $D_2O$, and measuring increasing fractions of produced $CH_3OD$ in a 1:1 correspondence [34].

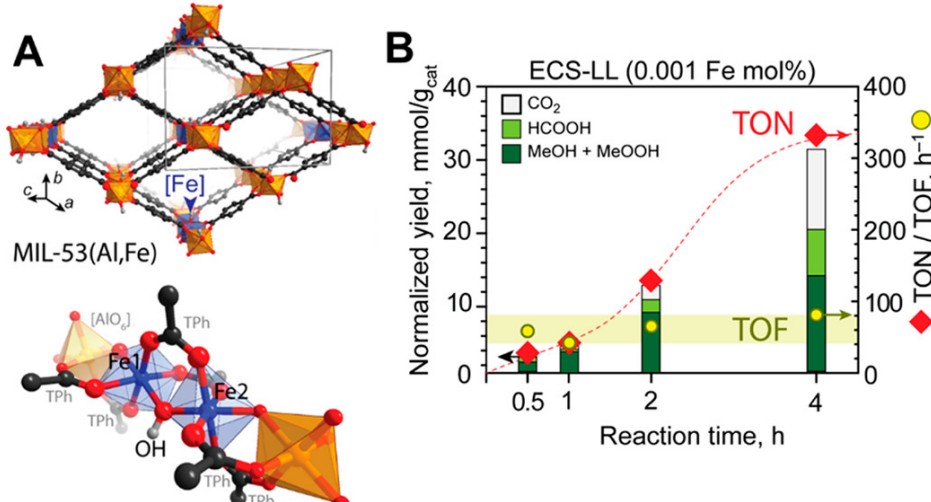

**Figure 4.** (**A**) Depiction of the MIL-53(Al,Fe) crystal structure, along with a representation of an isolated Fe-Fe site within the MOF octahedral $[AlO_6]$ chain. (**B**) Evolution of the oxidation products and the associated turnover frequency (TOF) and turnover number (TON) values during the MTM conversion in the presence of low concentrations of the ECS-LL catalyst (adapted with permission from Ref. [86]. Copyright 2018, American Chemical Society).

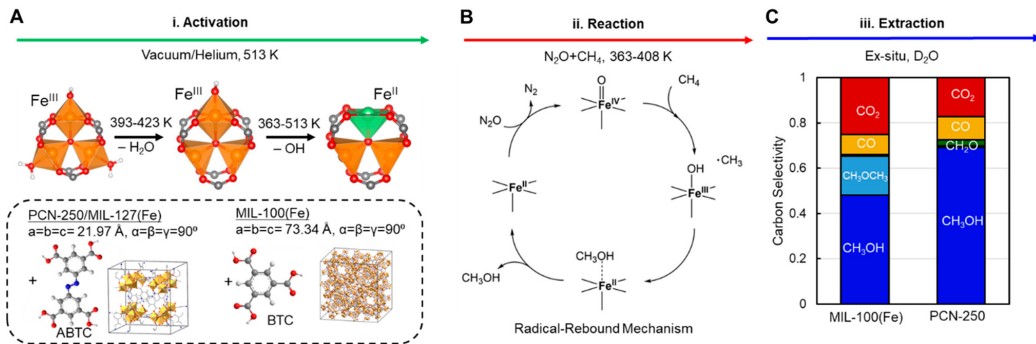

**Figure 5.** (**A**) Molecular structure of the $Fe_3$-$\mu_3$-oxo secondary building unit and its ligand modifications upon thermal activation. The unit cells of PCN-250 and MIL-100(Fe) are displayed in the inset, along with the corresponding ABTC and BTC linkers. (**B**) Schematic representation of the commonly proposed radical-rebound mechanism followed by methane activation on mononuclear Fe(II) centers exposed to mixtures of $N_2O$ + $CH_4$. (**C**) Cumulative carbon selectivity of products observed upon exposure of MIL-100(Fe)/PCN-250 to 80 kPa of $N_2O$, 10 kPa of $CH_4$, and 10 kPa of Ar at 120 °C for 4 h. The product selectivity is reported in ascending order: $CH_3OH$ (dark blue), $CH_3OCH_3$ (light blue), $CH_2O$ (green), CO (yellow), and $CO_2$ (red) (reprinted with permission from Ref. [66]. Copyright 2021, American Chemical Society).

In 2021, Simons and coworkers [66] disclosed that PCN-250, a MOF also built from $Fe_3$-$\mu_3$-oxo nodes incorporating a maximum of one high-spin (S=2) Fe(II) site per node (see Figure 5A), catalyzes the $N_2O$-based MTM reaction at temperatures inferior to 125 °C. Altogether, the Fe(II) sites arising from a thermal activation of the MOF initially form Fe(IV)=O centers by reacting with $N_2O$, while the methane homolytic activation was proposed to occur through a radical rebound mechanism often invoked when mononuclear Fe(II) sites are exposed to mixtures of $N_2O$ + $CH_4$ (Figure 5B). However, the findings that (i) the $N_2O$ + $CH_4$ reaction on PCN-250 yields various overoxidation products, such as $CH_2O$, CO, $CO_2$, and $CH_3OCH_3$ (see Figure 5C), and that (ii) the methanol product is observed

solely when washing the material ex situ with water, led the authors to assume underlying pathways of methanol product protection, i.e., invoking reaction steps beyond the radical rebound mechanism. On the basis of in situ IR measurements and DFT calculations, the authors proposed that methanol is stabilized within the MOF during the MTM reaction as methoxy groups located on the Fe-based SBUs. Furthermore, a composite constituted by the MOF and the MFI zeolite was exploited to improve methanol selectivity, yielding an amount of methanol equal to ca. 210 $\mu$mol $g_{cat}^{-1}$ at 1.1 bar and temperatures between 105 and 135 °C using a recirculating batch reactor [66].

More recently, in 2022, An and coworkers [68] reported that mono-iron hydroxyl sites immobilized in a Ru- and Fe-based MOF, termed PMOF-RuFe(OH), photocatalytically convert methane to methanol in the presence of $H_2O$ and $O_2$ with 100% selectivity. Here, we observe that the use of $O_2$ is particularly relevant, as molecular oxygen is more environmentally benign and less expensive than other common oxidants such as $H_2O_2$ or $N_2O$. The employed photocatalytic system integrated four different components: (i) a MOF based on the UiO-67 architecture to serve as a support platform; (ii) the photosensitizer [Ru(II)(bpy)$_2$(bpydc)] (bpy = 2,2'-bipyridine; H$_2$bpydc = 2,2'-bipyridine-5,5'-dicarboxylic acid) to absorb light; (iii) a polyvanadotungstate [PW$_9$V$_3$O$_{40}$]$^{6-}$ to activate $O_2$; (iv) a mono-iron hydroxyl site to activate methane [68]. The Ru photosensitizer and polyvanadotungstate moiety were introduced into the UiO-67 pores by a one-pot synthesis, while the Fe active sites were incorporated through a post-synthetic treatment using FeCl$_3$·6H$_2$O to yield PMOF-RuFe(Cl), which in turn afforded the active catalyst PMOF-RuFe(OH) upon pre-treatment in water with light irradiation for 2 h (the wavelength of light used was 400–780 nm). The presence of Fe–OH species was demonstrated by combining a series of different experimental techniques, including inelastic neutron scattering (INS), X-ray photoelectron spectroscopy (XPS), electron paramagnetic resonance (EPR), and X-ray absorption spectroscopy (XAS), supported by DFT calculations (refer to Section 5 for a more detailed discussion). In order to perform the MTM reaction, a continuous-flow photo-oxidation setup was implemented, where a packed layer of PMOF-RuFe(OH) catalyst was exposed to a flow of CH$_4$/O$_2$-saturated water under irradiation. The dynamic gas/solid/liquid interface maximizes contact between CH$_4$, O$_2$, H$_2$O, and the MOF and, notably, leads to a methanol time yield of 8.81 ± 0.34 mmol $g_{cat}^{-1}$h$^{-1}$ under ambient conditions, even outperforming the *Methylococcus capsulatus* (Bath) MMO (5.05 · mmol· $g_{cat}^{-1}$h$^{-1}$) [90].

Finally, a very recent study by Rungtaweevoranit and coworkers [91] reported a MOF with isolated Fe sites supported on the Zr-based UiO-66 structure, which may perform the MTM reaction under continuous gas-phase flow conditions in the presence of $O_2$ and water vapor. In the synthesized UiO-66 material there is approximately 1 acetate molecule and one H$_2$O/HO$^-$ pair per Zr$_6$ oxide cluster replacing the linker in missing-linker defects, which were leveraged to anchor Fe atoms onto the MOF. The activation of the Fe/UiO-66 catalyst was conducted by heating up to 250 °C under Ar and exposing the MOF to (i) 10% O$_2$/Ar at 250 °C for 1 h, (ii) 5% water vapor in Ar for 1 h at 250 °C, 1 bar, and (iii) Ar, while decreasing the temperature to a final value of 180 °C. The Fe/UiO-66 MOF was then exposed to the reaction gas mixture (10% CH$_4$, 5% O$_2$ + 0.2% H$_2$O, balance Ar) at 5 bar. During the first 250 min of the reaction, the rate of methanol formation increased up to $16.7 \times 10^{-2}$ $\mu$mol$_{methanol}$ $g_{Fe}^{-1}$ s$^{-1}$ (Figure 6A). Subsequently, the MOF catalyst entered a deactivation phase over 850 min, while increasing the reaction temperature to 190 °C resulted in a short increase of catalytic activity up to $10.3 \times 10^{-2}$ $\mu$mol$_{methanol}$ $g_{Fe}^{-1}$ s$^{-1}$ (see Figure 6A). Conversely, CO$_2$ formation reached a rate of $9.5^{-2}$ $\mu$mol$_{CO_2}$ $g_{Fe}^{-1}$ s$^{-1}$ over 650 min and was not affected by the temperature increase to 190 °C. Under the steady-state conditions at 180 °C (Figure 6B), the Fe/UiO-66 catalytic platform exhibited a selectivity towards the formation of methanol of 62%.

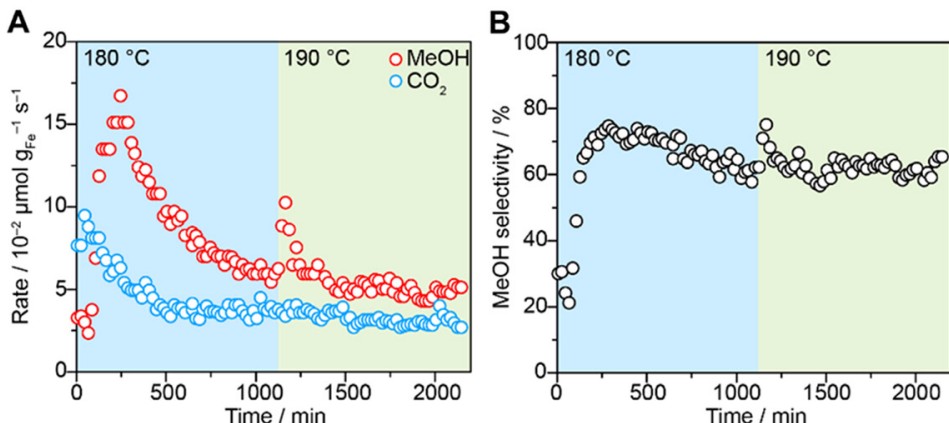

**Figure 6.** On-stream MTM conversion conducted over the Fe/UiO-66 MOF. (**A**) Rates of $CH_3OH$ and $CO_2$ formation during the direct $CH_4$ activation (10% $CH_4$, 5% $O_2$ + 0.2% $H_2O$, balance Ar) at 5 bar and at 180 or 190 °C. (**B**) Product selectivity towards the formation of methanol, expressed as $[CH_3OH]/([CH_3OH] + [CO_2])$ (adapted with permission from Ref. [91]. Copyright 2023, American Chemical Society).

The methane hydroxylation performance of the Fe-containing MOFs illustrated in this section is summarized in Table 1.

**Table 1.** Summary of the MTM conversion performance for the Fe-containing MOFs illustrated in Section 3. * depending on Fe wt %.

| MOF | Active Site Type | Oxidant | Methanol Yield | Methanol Selectivity | Reference |
|---|---|---|---|---|---|
| MIL-53(Al,Fe) | FeO$_6$ SBU | $H_2O_2$ | $\sim$10–30 $\mu molg_{cat}^{-1}$ * | $\sim$10–40 * | [86] |
| MIL-100(Fe) | Fe$_3$-$\mu_3$-oxo trimer | $N_2O$ | 0.34 mol(mol Fe)$^{-1}$ | $\sim$99% at 453 K | [34] |
| PCN-250 | Fe$_3$-$\mu_3$-oxo trimer | $N_2O$ | 200 $\mu molg_{cat}^{-1}$ | 70% | [66] |
| PMOF-RuFe(OH) | mono-iron hydroxyl | $O_2$ | $8.81 \pm 0.34$ mmolg$_{cat}^{-1}$h$^{-1}$ | 100% | [68] |
| Fe-UiO-66 | mono-iron hydroxyl | $O_2$ | 0.167 $\mu molg_{cat}^{-1}s^{-1}$ | 62% | [91] |

## 4. MOF-Supported Cu Clusters for the MTM Conversion

Particulate methane monooxygenase (pMMO) is a copper-based membrane enzyme, expressed by nearly all methanotrophic organisms, and serves as the insoluble counterpart of the sMMO enzyme [92]. Organisms that may express both sMMO and pMMO exhibit a preference for pMMO expression in conditions of high copper availability [93,94], despite the better performance of sMMO in terms of methanol turnover frequency and catalytic efficiency [95].

However, unlike sMMO, many properties of pMMO are not yet fully understood due to the low resolution of the available enzyme crystal structures (<2.68 Å [96]). For instance, the exact structure of the copper active site(s) is yet to be disclosed [97], and evidence supporting the presence of both monocopper and dicopper sites has been presented often proposing their first-shell coordination with histidine residues [96,98]. As an example, Figure 7 shows putative monocopper and dicopper pMMO active site structures optimized through hybrid quantum mechanics/molecular mechanics (QM/MM) methods [99]. This degree of uncertainty regarding the pMMO active site, as well as the absence of dicopper sites in similar enzymes with well-determined structures, have hindered the full comprehension of the pMMO-mediated MTM reaction mechanism.

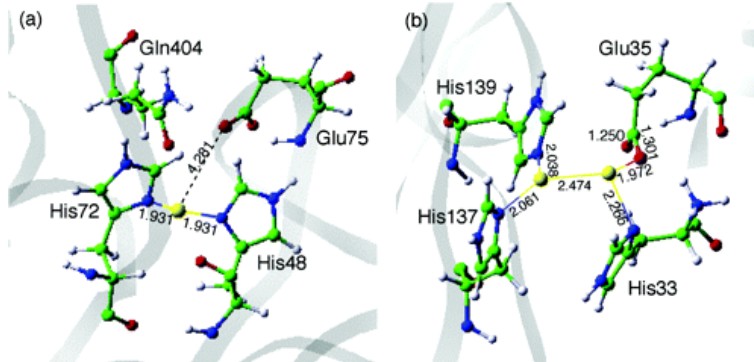

**Figure 7.** QM/MM structures of the monomeric (**a**) and dimeric (**b**) copper active sites of pMMO optimized from crystallographic data (adapted with permission from Ref. [99]. Copyright 2006, American Chemical Society).

Nonetheless, numerous reaction schemes have been proposed to explain the reactivity of pMMO during the MTM process. A DFT-based study by Yoshizawa and coworkers modeled the reaction pathway over a dicopper site (see Figure 8), assuming a radical rebound mechanism [99]. Following $O_2$ activation, a superoxo-bridged doublet dicopper active species ($^2$Oxo) is formed. This species interacts with methane leading to intermediate $^2$R, which subsequently undergoes an H-bond cleavage process through a transition state $^2$TS (18.9 kcal/mol) to form the methyl radical ($^2$Int). Then, a radical rebound step ($^2$TS2, 25.9 kcal/mol) finally yields $CH_3OH$ and a monoxygenated dicopper site $^2$P while releasing 45.1 kcal/mol.

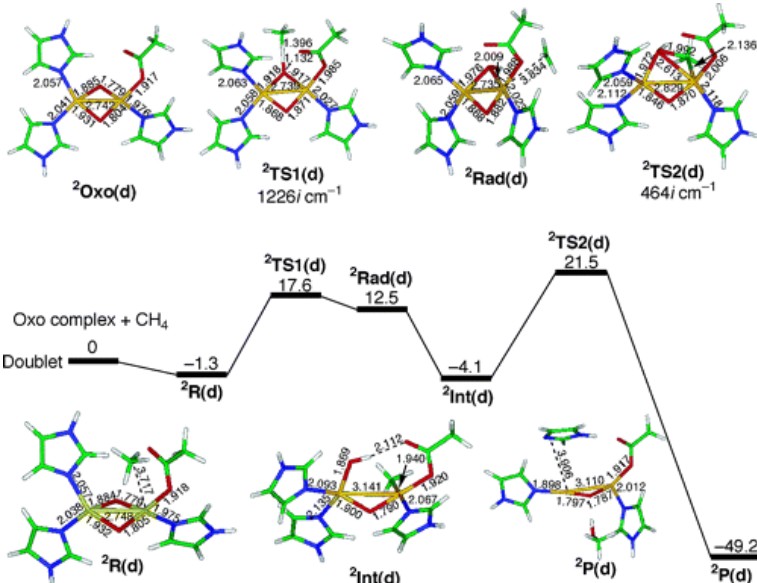

**Figure 8.** Energy diagram for the MTM conversion over a proposed pMMO dimeric copper active site (energy units are expressed in kcal/mol) (adapted with permission from Ref. [99]. Copyright 2006, American Chemical Society).

Many attempts at replicating the reactivity of pMMO have been carried out, starting from model dicopper complexes that can oxidize weak C–H bonds [100]. Similarly, Cu-exchanged zeolites that exhibit mononuclear, dinuclear, and even trinuclear copper active sites have been studied for the MTM conversion showing good yields [101–103].

At variance with the inorganic skeleton of zeolites, the hybrid MOF structures are believed to more closely resemble the environments offered by polypeptide chains in enzymes [69], enabling one to pursue biomimetic reactivity while employing MOF catalysts. Following this principle, in 2018, Baek and coworkers installed three different

imidazole-based ligands (5-benzimidazolecarboxylate, L-histidine, and 4-imidazole acrylate) on the Zr-based MOF-808 framework [69], subsequently loading Cu(I) centers on the ligands to form dicopper sites almost identical to those proposed in pMMO (see Figure 9). Although the structure of these slightly disordered dimeric sites is elusive to crystallographic methods, their presence was confirmed via energy-dispersive X-ray spectroscopy (EDS), N K-edge and Cu K-edge XAS, UV–Vis diffuse reflectance spectroscopy (DRS), as well as by resonance Raman spectroscopy measurements. Catalytic tests showed that these copper sites effectively oxidize methanol at 150 °C using $N_2O$ as the oxidant, with a methanol productivity of up to $71.8 \pm 23.4$ µmol $g_{cat}^{-1}$ when 5-benzimidazolecarboxylate acts as the support (MOF-808-Bzz-Cu). The other two imidazole ligands showed reduced activity, with L-histidine (MOF-808-His-Cu) displaying the lowest methanol productivity of $31.7 \pm 13.0$ µmol $g_{cat}^{-1}$. Notably, only methanol and water were observed as reaction products during methanol steam desorption experiments, although a certain amount of $CO_2$ was formed as a reaction byproduct. The recyclability of MOF-808-Bzz-Cu was also explored, and subsequent reaction cycles over the catalyst showed a large degree of deactivation with methanol productivity being reduced from 71.8 to 7.5 µmol $g_{cat}^{-1}$ between the first and the second reaction cycle. This effect was attributed to strongly adsorbed water molecules that hamper the catalytic process [69]. Powder X-ray diffraction (PXRD) analyses evidenced the structural stability of all three MOF catalysts under reactive conditions.

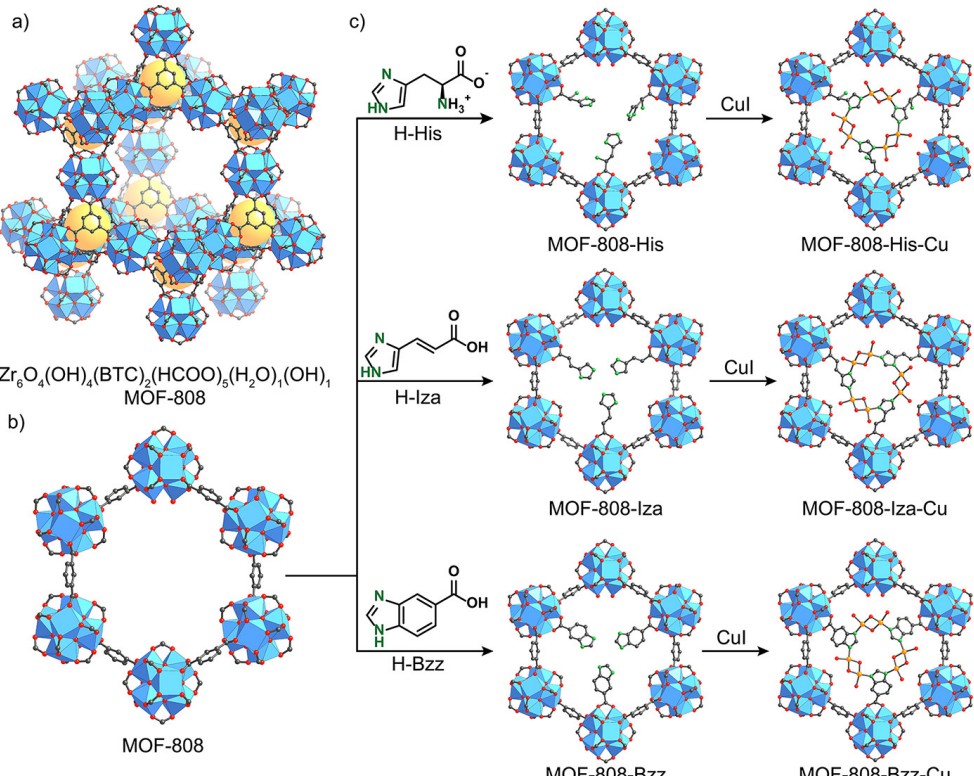

**Figure 9.** Synthesis of bioinspired MOF-808-supported Cu catalysts. (**a**) MOF-808 structure. (**b**) Pseudohexagonal pore opening of MOF-808. (**c**) Replacement of formate groups with imidazole-containing ligands and subsequent metalation with Cu(I) (adapted with permission from Ref. [69]. Copyright 2018, American Chemical Society).

Different strategies have also been explored to realize Cu-loaded Zr-based MOFs. In 2017, Ikuno and coworkers inserted oligomeric copper-oxo clusters within the NU-1000 MOF (see Figure 10) using the atomic layer deposition (ALD) technique [71]. In this way, a 10 % wt copper-loaded Cu-NU-1000 MOF was obtained, possessing an average of 4 copper atoms per node and a crystal structure identical to that of its pristine counterpart, as confirmed by PXRD and high-angle annular dark-field scanning transmission electron

microscopy (HAADF-STEM) analyses. The structure and oxidation state of the copper-oxo clusters, whose proposed DFT-derived geometry is depicted in Figure 11A, were investigated using XAS, XPS, and pair distribution function (PDF) measurements, and closely resemble those of copper sites present in Cu-NU-1000 and $Cu(OH)_2$ [104]. Most of the copper species (~85%) were determined to be in an octahedrally coordinated Cu(II) state, while ~15% of Cu(I) sites in two- or three-coordinated geometries were detected. Several MTM reaction conditions were explored for the $O_2$-based MTM conversion at 150 °C over this MOF, following pretreatment in $O_2$ flow at 200 °C for 3 h to remove physisorbed water. When exposing the pre-treated Cu-NU-1000 to a $CH_4$ flow at 150 °C for 3 h and then to a 10% steam He flux at 135 °C, methanol was produced with a yield of 17.7 $\mu$mol $g_{cat}^{-1}$. Dimethyl ether (DME), a valuable oxygenate of high industrial relevance, was also detected in small quantities (2.0 $\mu$mol $g_{cat}^{-1}$), for a total carbon selectivity (methanol + DME) of ~45%. Two subsequent reaction cycles were performed, showing reduced reactivity with yields of 15.8 and 13.2 $\mu$mol $g_{cat}^{-1}$, respectively. When increasing the water content of the steam He flow to 50%, methanol yield and carbon selectivity were both drastically reduced (6.9 $\mu$mol $g_{cat}^{-1}$ and ~14%). This was speculated to be due to the decarboxylation of the Cu-NU-1000 linkers, as 60% MOF pore volume was lost while a reduction of only 1–2% was observed while exposing the system to a 10% water He flux.

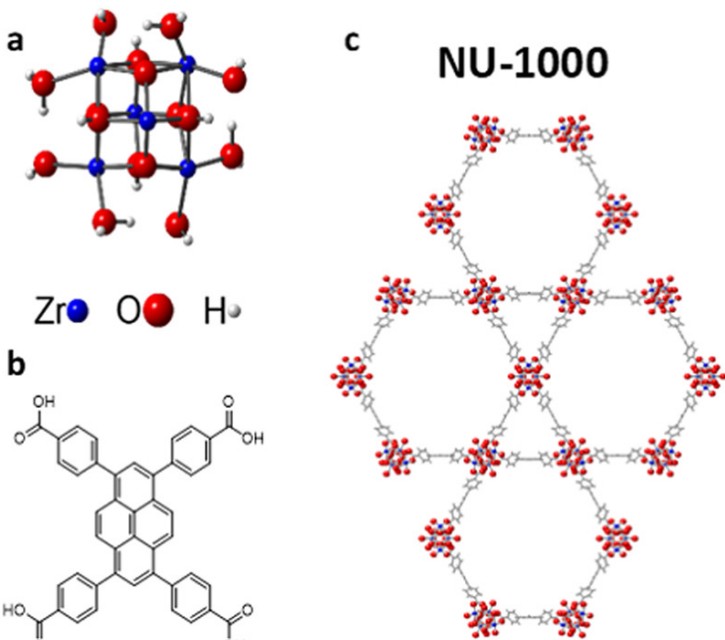

**Figure 10.** Depiction of the NU-1000 (**a**) hexa-Zr nodes and (**b**) tetratopic organic linkers, as well as of the (**c**) MOF 3 nm-wide hexagonal pores (reprinted with permission from Ref. [105]. Copyright 2020, American Chemical Society).

In 2019, Zheng and coworkers also exploited NU-1000 as a platform for copper ion loading through various degrees of cation exchange over the terminal -OH groups present in the zirconium oxide SBUs [70]. Three different copper loadings were explored, namely 2.9 wt % (Cu-2.9-NU-1000), 1.9 wt % (Cu-1.9-NU-1000), and 0.6 wt % (Cu-0.6-NU-1000), and also in this case the obtained MOFs were found to be isostructural to the pristine Zr-based one. Scanning transmission electron microscope-energy dispersive X-ray spectroscopy (STEM-EDS) measurements evidenced a uniform distribution of copper sites. Extended X-ray absorption fine structure (EXAFS) measurements were leveraged to probe their local structure, which was again found to be nearly identical to that of copper within $Cu(OH)_2$, with almost all copper species displaying the +2 oxidation state in monomeric (prevalent in Cu-0.6-NU-1000) or dimeric units (prevalent in Cu-2.9-NU-1000), as shown in Figure 11B. Furthermore, the methanol yield was demonstrated to correlate with the copper loading. In

particular, the highest yield (4.4 μmol $g_{cat}^{-1}$) was obtained while using Cu-2.9-NU-1000, and after activating the catalyst with 1 bar $O_2$ at 200 °C and exposing it to 1 bar $CH_4$ at 150 °C. Furthermore, Cu-2.9-NU-1000 produced approximately 0.01 methanol molecules per single copper site, a degree of activity comparable to that of the ALD-synthesized Cu-NU-1000 MOF (*vide supra*) and higher than that obtained with copper-exchanged zeolites such as Cu-MOR or Cu-ZSM [106,107]. Moreover, although the previously reported ALD-prepared Cu-NU-1000 outperforms Cu-2.9-NU-1000 in terms of methanol yield, the latter can be more readily prepared and displays a very high methanol selectivity of 70%. Remarkably, recyclability tests over five consecutive cycles also indicated only slight catalyst deactivation after the first cycle with stable methanol selectivity, and DME was not observed as a byproduct. Methanol selectivity was confirmed through isotope labeling experiments that made use of $^{13}$C-methane as the reactant under the same conditions: $^{13}$C-labeled methanol and $^{13}$C-labeled $CO_2$ were found to be the main products with only minor amounts of $^{12}CO_2$ being released. Increasing the methane pressure to 40 bar during the reaction step resulted in increased methanol productivity by a factor of ∼3.5, together with the formation of DME in small quantities. Conversely, the increase in methanol overoxidation to $CO_2$ was negligible and the total carbon selectivity at 150 °C and 40 bar $CH_4$ was found to be as high as ∼90% (with 0.04 methanol molecules produced per copper site). Since the methanol yield in dicopper-dominated Cu-2.9-NU-1000 was more than six times higher than that of monocopper-dominated Cu-0.6-NU-1000, it was concluded that dimeric copper oxyl species were responsible for most of the MTM reactivity over Cu-exchanged NU-1000. The MTM reaction mechanism was investigated with DFT calculations, which indicated that the reaction initially proceeds through a homolytic C–H bond dissociation over $Cu(II)_2(OH)_4$ clusters, producing a $CH_3^{\bullet}$ radical and an $H_2O$ ligand (21.2 kcal/mol). In turn, the methyl radical then abstracts an $HO^{\bullet}$ radical from the $H_2O$ ligand to form methanol (14.7 kcal/mol). DFT modeling also predicted the formation of $Cu(II)_2(O^{\bullet})(OH)_3$ copper-oxyl species in relatively small quantities. These species were also predicted to hydroxylate methane following a mechanism based on H-atom abstraction over the oxyl moiety while requiring a significantly smaller activation energy (18.9 kcal/mol) if compared to that required by the reaction occurring over $Cu(II)_2(OH)_4$ sites (35.3 kcal/mol). This result suggested that the very low MTM reactivity of Cu-2.9-NU-1000 is due to the fact that in this MOF copper oxyl clusters, the most active species are only present in very limited quantities.

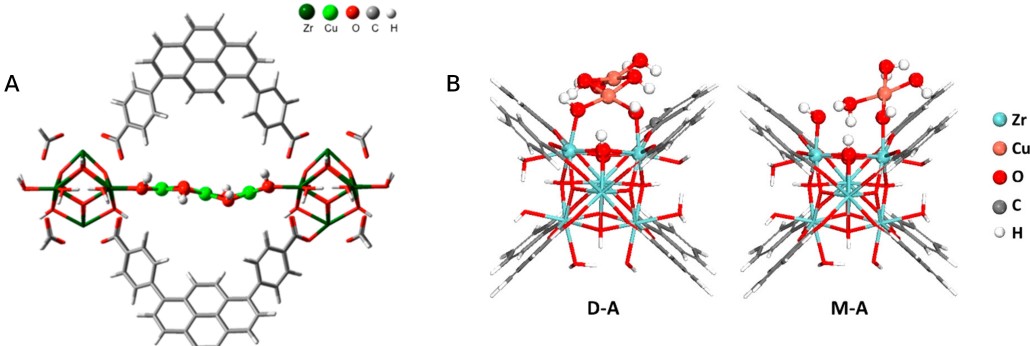

**Figure 11.** Proposed DFT structures of the Cu-NU-1000 active sites, synthesized either by atomic layer deposition (**A**) or by metal exchange strategies (**B**). ((**A**) Reprinted with permission from Ref. [71]. Copyright 2017, American Chemical Society). ((**B**) Reprinted with permission from Ref. [70]. Copyright 2019, American Chemical Society).

In 2021, Ren and coworkers exploited a ligand-coordination strategy to insert square-planar $CuCl_2$ clusters coordinated by the 2,2′-bipyridine-5,5′-dicarboxylate (bpydc) ligand within the UiO-bpy MOF [108]. Such Cu-based clusters have been shown to produce linearly coordinated $Cu_xO_y$ species upon activation in $O_2$ flow at 200 °C [109], and XPS measurements indicated that their structure and oxidation state were very close to those observed in CuO. The MTM reaction was then carried out on the activated system and

involved isothermal $CH_4$ loading for 3h and methanol desorption by exposing the system to a steam/He flux. The reported methanol yield was equal to 24.33 $\mu$mol $g_{cat}^{-1}$, with a selectivity of 88.1% during the first reaction cycle. Ethanol was formed as the main reaction byproduct while no amount of $CO_2$ was detected in these conditions. During subsequent reaction cycles, the methanol yield was only slightly reduced while selectivity reached 100% as no byproducts were detected, indicating optimal catalyst recyclability. Thermodynamic calculations predicted that two-dimensional $Cu_4O_4$ species form in the MOF channels, and over these Cu sites the DFT absorption energy of methane was found to be as low as $\sim-0.9$ kcal/mol. A DFT-based MTM reaction mechanism was also reported, according to which the interaction between $CH_4$ and two distinct oxygen sites leads to the formation of $CH^\bullet$ and an OH site with an energy barrier of $\sim$27 kcal/mol. This step is followed by the exothermic production of methanol with a relatively low activation energy of $\sim$6 kcal/mol.

Lastly, in 2022, Lee and coworkers demonstrated that the copper-doped ZIF-7 (zinc benzimidazolate) system can hydroxylate methanol using $H_2O_2$ as the oxidant [110]. PXRD analysis evidenced a phase transition of the catalyst to a square-planar dense structure upon copper insertion, while XAS analyses disclosed the presence of four-coordinated Cu(II) species. The reaction of methane with Cu/ZIF-7 and $H_2O_2$ (0.5 M) in an aqueous solution produced methanol as well as methyl hydroperoxide, hydroxymethyl hydroperoxide, and carbon dioxide as overoxidation products, while a blank test performed with pristine ZIF-7 only yielded formic acid due to the presence of strongly acidic zinc sites. Through isotopic labeling catalytic tests, undesired oxidation processes of the benzimidazole ligands were identified as the source of the carbon dioxide product. The influence of reaction conditions was investigated, showing that $CH_4$ can already be activated at 30 °C. Furthermore, it was found that the total yields of methanol and its overoxidation products increased as a function of temperature until a drastic reduction in selectivity was observed at 90 °C, while the maximum-yield temperature (70 °C) caused a deterioration of the framework. The catalyst performance was, therefore, determined to be optimal at 50 °C, with an elevated methanol yield of 48 $\mu$mol $g_{cat}^{-1}$. Increasing the concentration of hydrogen peroxide above 0.5 M increased the production of hydroxymethyl hydroperoxide and $CO_2$ due to the augmented methanol overoxidation and ligand oxidation. To assess the reusability of the catalyst, the MTM process was repeated five times under optimized conditions demonstrating unaffected reactivity. Furthermore, although the catalyst weight gradually decreased after each use, XAS and PXRD analyses showed no structural and electronic modifications of the Cu/ZIF-7 system. The reaction mechanisms leading to the formation of both methanol and methyl hydroperoxide were studied through DFT calculations. Specifically, it was proposed that an initial dissociative reaction of $H_2O_2$ to form $H_2O$ and $O\text{-}CuN_4$ as products occurs after the adsorption of $HOO^\bullet$ on the given copper site. A copper-oxyl species was then predicted to hydroxylate methane in an H-bond cleavage step, followed by radical rebound-driven methanol formation as observed in several iron- and copper-based systems [66,99]. The activation barrier for this step was determined to be $\sim$15.2 kcal/mol. Importantly, it was found that the reaction of $O_2$ with the methyl radical produces $CH_3OO$, a stable intermediate that may abstract hydrogen from hydroxylated $HO\text{-}CuN_4$ sites to form methyl hydroperoxide.

Table 2 summarizes the methane hydroxylation performance of the Cu-bearing MOFs illustrated in this section.

**Table 2.** Summary of the MTM conversion performance for the Cu-containing MOFs illustrated in Section 4.

| MOF | Active Site Type | Oxidant | Methanol Yield ($\mu mol g_{cat}^{-1}$) | Methanol Selectivity | Reference |
|---|---|---|---|---|---|
| MOF-808-Bzz-Cu | histidine-supported Cu-oxo dimer | $N_2O$ | $71.8 \pm 23.4$ | 100% | [69] |
| Cu-NU-1000 | Cu-oxo cluster | $O_2$ | 17.7 | 45% (including DME) | [71] |
| Cu-2.9-NU-1000 | Cu-oxo dimer | $O_2$ | 4.4 | 70% | [70] |
| Cu-UiO-bpy | $Cu_xO_y$ clusters | $O_2$ | 24.3 | 88% | [108] |
| Cu-ZIF-7 | ligand-supported tetrahedral Cu | $H_2O_2$ | 48 | 8% | [110] |

## 5. Spectroscopic Characterization of the MOF-Based Methane Oxidation

To begin this section, we observe that the development of next-generation catalysts to accomplish the efficient and selective MTM conversion is a highly complex and challenging process, where basic science, screening methodologies, and engineering are all required to contribute in a synergic manner [111]. Nonetheless, while the rate and selectivity of the MTM reaction are the most sensitive descriptors of a given catalyst's performance, these macroscopic criteria are not sufficient to deduce the underlying reaction mechanism and need to be complemented with additional microscopic information on the nature, structure, electronic properties, and reactivity of the key reaction active sites and intermediates. It is for such information that one resorts to spectroscopic techniques, which, when complemented with theoretical methods, may shed light on the active reaction mechanism and provide insights that are key to rationally designing improved catalysts [111]. When considering the MTM reaction performed over iron and copper sites in MOFs, our attention primarily focuses on spectroscopic tools useful in heterogeneous catalysis, where surface properties are of great significance to drive the desired reaction pathways. In fact, the permanent porosity and very high surface areas exhibited by MOFs render these materials as unique platforms for solid-gas catalytic reactions. The "ideal" spectroscopic technique to investigate the MTM reaction over MOFs would allow the identification of adsorbed molecules, while providing information on the structure, oxidation, and spin states of the surface species, and on the intermolecular interactions established by adsorbed species (such as $CH_4$ itself and the $H_2O_2$, $N_2O$ or $O_2$ oxidants) with the framework atoms. Furthermore, the measurements should be possible over a wide range of pressures and temperatures. While such an "ideal" spectroscopic probe does not exist as a single method alone, the judicious combination of diverse and complementary spectroscopic techniques operating in different energy ranges of the electromagnetic spectrum may often answer the majority of the experimental questions [111].

Some properties of individual spectroscopies relevant to heterogeneous catalysis and employed to investigate the MTM reaction over MOFs are summarized in Table 3. It is worth observing that for all spectroscopies there are restrictions on the nature and form of the investigated sample, as well as on the experimental conditions allowed by the given technique. For instance, ambient pressures are not accessible for XPS, because the mean free path of electrons is very limited even at pressures inferior to $10^{-2}$ atm, while EPR and Mössbauer spectroscopies require, respectively, unpaired electrons and Mössbauer-active nuclei, such as the $^{57}$Fe isotope.

**Table 3.** Comparison of different kinds of spectroscopic techniques that are useful for investigations in heterogeneous catalysis.

| Spectroscopy | Energy (eV) | Kind of Transition | Information Mainly Provided on: | | |
|---|---|---|---|---|---|
| | | | Solid Surface | Bond between Adsorbed Molecule and Surface | Adsorbed Molecule Structure |
| Infrared | $(50–2.5) \times 10^{-2}$ | Vibrational | N | Y | Y |
| Raman | $(50–0.6) \times 10^{-2}$ | Vibrational | N | Y | Y |
| Visible/near infrared | 0.5–6.5 | Electronic, vibrational | Y | Y | Y |
| Mössbauer | $10^4–10^5$ | Nuclear | Y | N | N |
| EPR | $(14–3.8) \times 10^{-5}$ | Nuclear spin | Y | N | Y |
| NMR | $(2.5–8.3) \times 10^{-7}$ | Nuclear spin | Y | Y | Y |
| XPS | (0.1–1500) | Bound electron to the continuum | Y | Y | Y |
| XAS | $(0.1–100) \times 10^3$ | Bound electron to the continuum | Y | Y | Y |

As reported in Table 3, IR, Raman, NMR, and EPR spectroscopies are more frequently employed to investigate the structural properties within molecules adsorbed on surfaces, offering also the possibility of making deductions about the adsorption configuration. Conversely, optical, Mössbauer, and XPS techniques furnish direct insights into surface composition and on the local structural and electronic properties of the adsorption sites. Notably, XAS measurements in both the soft (0.1–3 keV) [112–115] and hard (3–100 keV) [116–122] energy ranges may provide—at the same time—information on the nature, structure, and time evolution of the key intermediates with element-specificity and high sensitivity. It should be noted that the majority of works reported in the literature employed a combination of the above-mentioned spectroscopic techniques, supported by DFT calculations, to characterize the catalytic active sites and the intermediate species formed while performing the MTM conversion over iron- and copper-bearing MOFs.

For instance, Gascon and coworkers [86] utilized Mössbauer, EPR, and XAS spectroscopies to characterize the Fe species present in the MIL-53(Al,Fe) catalysts. From the EPR and Mössbauer experiments, it was found that while the HTS samples mostly contain oligomeric species, the degree of dispersion observed in the ECS samples was higher. In particular, for the ECS samples, the Mössbauer spectra recorded at $-231.15$ °C (see Figure 12A) display a dominant doublet component with no magnetic hyperfine structure alongside a small component related to single Fe(III) atoms, supporting the view that the catalysts are comprised of a combination of isolated monomeric Fe(III) centers and antiferromagnetically coupled dimeric Fe(III)–Fe(III) sites [86]. Fe K-edge X-ray absorption near edge structure (XANES) spectra evidenced that octahedrally coordinated Fe(III) sites are present in all samples while an EXAFS analysis (Figure 12B) evidenced that each iron center is directly coordinated by two O atoms at 1.95–1.96 Å, which are associated with the shorter bridging Fe–$\mu$-O–Fe bonds, and two O atoms at 2.01–2.03 Å.

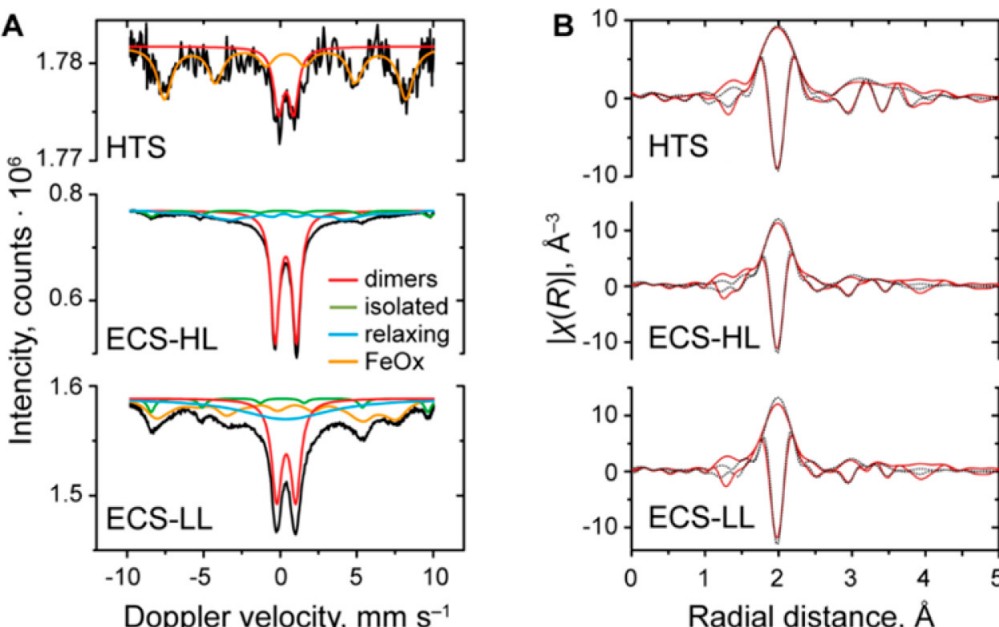

**Figure 12.** (**A**) Mössbauer spectra of the HTS, ECS-HL, and ECS-LL catalysts in zero field at −231.15 °C (the experimental data are represented by black solid lines, and the corresponding fitted components are represented by colored lines). (**B**) Fourier transform of EXAFS Fe K-edge experimental (red) and fitted (black) data for the HTS-1, ECS-HL, and ECS-LL samples (adapted with permission from Ref. [86]. Copyright 2018, American Chemical Society).

Bollini and coworkers [34] employed IR spectroscopy to demonstrate that solely Fe(II) centers are active towards the MTM conversion over MIL-100(Fe), using NO as a probe molecule selective to the MOF Fe(II) active centers. In fact, at sufficiently low NO pressures (below 5 kPa NO) Fe(II) open metal sites in MIL-100(Fe) are saturated by NO with no indication of Fe(III)-nitrosyl species being formed [64]. Specifically, exposing the activated MIL-100 material to 0.5 kPa NO at 150 °C resulted in a band at 1810 cm$^{-1}$ assigned to NO directly coordinated with the MOF Fe(II) sites. Furthermore, the (Fe(II)–NO) and (Fe–OH) IR relative band areas were found to correlate linearly. This evidence supported the view that thermally treating the MOF to progressively higher temperatures generates larger Fe(II) site densities, with near complete dehydroxylation achieved at 250 °C where one Fe(II) site per SBU is obtained.

Simons and coworkers [66] employed in situ XAS to track the structural and electronic variations of the Fe sites in PCN-250 both after thermal activation and after the $N_2O + CH_4$ reaction. Figure 13A compares the XANES of the as-synthesized (black), thermally activated (red), and post-reaction (blue) materials. Overall, thermally activating the MOF in He to 240 °C produced a shift of the XANES absorption edge energy to lower energies as well as a decrease in the white line intensity. In addition, a decrease in the Fourier transform (FT) magnitude of the EXAFS spectrum at 2.03 Å was observed during activation (see Figure 13B), in agreement with the temperature-induced removal of $H_2O$ and $HO^-$ species directly coordinating the iron MOF centers. After performing the MTM reaction at 120 °C, the XANES absorption edge shifted once again to higher energies while the magnitude of the FT at 2.04 Å increased, indicating that Fe(II) sites had been converted back to Fe(III) and likely coordinated by additional ligands. Fitting of the EXAFS data based on structural models in line with the XRD structure of PCN-250 [123] evidenced that activation from 125 to 240 °C only negligibly affected the average Fe–O coordination number, from 5.8 ± 0.5 to 5.6 ± 0.5, while in the post-reaction catalyst, the average Fe–O coordination number slightly increased again. This finding was consistent with the notion that apical $HO^-$ ligands had been removed during activation and replaced by other axial ligands after the MTM conversion. Furthermore, the authors resorted to in situ IR spectroscopy to investigate the mechanisms of $CH_3OH$ protection during MTM reactive conditions. In the

IR spectrum of PCN-250, activated by heating in vacuo to 240 °C, there are solely $\nu_{CH}$ bands belonging to the organic linker. Conversely, in the IR spectrum of the post-reaction MOF a $\nu_{OH}$ band at 3682 cm$^{-1}$ and three new $\nu_{CH}$ peaks appear (at 2904, 2876, and 2802 cm$^{-1}$) which correlate with methanol formation quantified by means of $^1$H-NMR spectroscopy after washing the IR sample with D$_2$O. These $\nu_{CH}$ bands are associated with specific C–H vibrations that belong to a methanol-derived species stabilized within the MOF pores and DFT calculations suggested they belong to a methoxy group coordinating a Fe(III) site within the PCN-250 SBU. The authors hypothesized that the formation of such stable methoxy species was due to CH$_3$OH molecules reacting with MOF surface hydroxyl groups (observed in PCN-250 using IR measurements) via water elimination. This was therefore proposed to constitute a pathway for the protection of gaseous CH$_3$OH additional to the radical rebound mechanism. In fact, exposing PCN-250 bearing Fe(III)–OH groups to gas-phase CH$_3$OH resulted in a decrease of the $\nu_{OH}$ band and in the formation of three new bands at 2904, 2876, and 2802 cm$^{-1}$ that were assigned to Fe(III)–OCH$_3$ species [66].

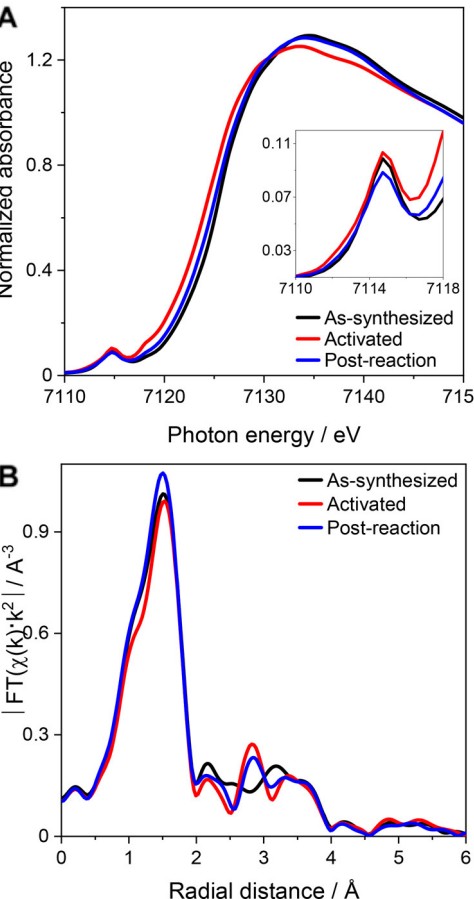

**Figure 13.** Fe K-edge XANES (**A**) and EXAFS (**B**) spectra (k$^2$-weighted magnitude of the FT) characterizing PCN-250 in flowing He at 120 °C (black curve), at 240 °C (red curve), and after performing the MTM reaction with N$_2$O + CH$_4$ at 120 °C (blue curve) (reprinted with permission from Ref. [66]. Copyright 2021, American Chemical Society).

An and coworkers [68] utilized a combination of diverse spectroscopic tools to establish the structure of the PMOF–RuFe(Cl) and PMOF–RuFe(OH) materials. PXRD patterns demonstrated that PMOF–RuFe(Cl) and PMOF–RuFe(OH) retain the UiO-67-type structure, while binding of Fe(III) ions to the MOF linker was supported by means of UV–Vis–near IR spectroscopy. XPS analyses of the active PMOF-RuFe(OH) species indicated that Fe(III)–OH species were present, Raman measurements excluded the formation of O–O bonds as in possible Fe–O$_2$ adducts [68], while EPR spectra showed that both PMOF–RuFe(Cl) and PMOF–RuFe(OH) contained high-spin Fe(III) sites with highly rhombic environments. The

XANES spectrum of PMOF–RuFe(OH) displays a 1s→3d transition at 7114.8 eV, which is less intense and at slightly higher energies if compared to the pre-edge peak in the XAS spectrum of the PMOF–RuFe(Cl) material (7114.4 eV). The EXAFS signal of PMOF–RuFe(OH) is consistent with a DFT-optimized $[(bpy)Fe(OH)(H_2O)_3]^{2+}$ complex, where the Fe–$O_{OH}$ and Fe–$O_{H_2O}$ bonds are equal to 1.78 ± 0.09 Å and 2.17 ± 0.09 Å, respectively [68]. Finally, a wavelet transform analysis of the EXAFS spectrum of PMOF–RuFe(OH) indicated that Fe···Fe binuclear or cluster species are not present, as they do not contribute to second-shell scattering if compared to a $Fe_2O_3$ reference which exhibits Fe–Fe distances of ~2.5 Å [68].

Rungtaweevoranit and coworkers [91] collected $^{57}$Fe Mössbauer spectra of the as-synthesized Fe/UiO-66 catalyst at −268.15 °C, and −193.15 °C to characterize the MOF iron sites. These measurements supported the presence of both $Fe_2O_3$ nanoparticles and high-spin Fe(III) centers octahedrally coordinated by oxygen atoms in an approximately 1:1 ratio. Furthermore, EPR analysis of the Fe/UiO-66 MOF identified two signals with *g* values of 2.01 and 4.24 attributed to $\gamma$–$Fe_2O_3$ and an isolated Fe(III) species in distorted octahedral coordination, respectively. The Fe K-edge XAS spectrum of Fe/UiO-66 possesses a pre-edge transition at 7114 eV, which is in the same region as that of $Fe_2O_3$, suggesting that the MOF iron sites have a +3 oxidation state, while EXAFS data fits indicated that Fe is 6-coordinated and bound to $HO^-$ and $H_2O$ ligands. In addition, the XANES edge transition is located at 7127 eV, a value higher than typical Fe(III)-related absorption edges, supporting the view that the Fe sites are electron-deficient [91]. After pre-treating the Fe/UiO-66 material and performing the MTM reaction, the iron first-shell coordination parameters remained basically unaffected while an increase in the Fe···Fe backscattering coordination number (CN) was observed, likely due to the agglomeration of Fe sites into $FeO_x$ nanoparticles. XPS analysis of the Fe 2p region indicated the presence of two components, the first one assigned to Fe(III) residing in $Fe_2O_3$ and the second one associated with the Fe(III) sites bound to electron-withdrawing ligands, a finding in agreement with the XAS data. To better track the evolution of the Fe sites during the MTM conversion, in situ DR UV–Vis spectroscopy was employed, evidencing that while the pre-reaction MOF samples contain a mixture of mono-, bi-, and polynuclear Fe species, after the MTM reaction these isolated Fe species agglomerate into larger $FeO_x$ nanoparticles. In situ IR measurements found that, while performing the MTM reaction, peaks arise, attributable to surface methoxy and formate-related species (e.g., formate, formaldehyde, and formic acid).

Ikuno and coworkers [71] performed Cu K-edge XAS measurements to determine the electronic and structural properties of the copper sites in the ALD-synthesized Cu-NU-1000 MOF. A linear combination fitting of the XANES spectrum of the pristine MOF (see Figure 14A) using $Cu_2O$ and $Cu(OH)_2$ as Cu(I) and Cu(II) references, respectively, suggested that approximately 15% of Cu is present as Cu(I) and ~85% as Cu(II). Furthermore, there were only minor variations in the XANES and EXAFS spectra while activating the MOF in flowing $O_2$ at 150 °C (Figure 14B), while an increase in the Cu(I) characteristic 1s→4p transition was observed when exposing the activated MOF to $CH_4$, with an estimated ~9% of Cu(I) being further reduced (Figure 14C). The evidence indicates that the majority of copper sites in the MOF retained the +2 oxidation state. The structural environment around the copper atoms was investigated by means of a Cu-EXAFS analysis. Specifically, the Cu–O bond was found to be shorter than in $Cu(OH)_2$ suggesting the coexistence of multiple copper species in the MOF. Furthermore, the EXAFS spectrum of Cu-Nu-1000 (Figure 14D, black curve) possesses a single scattering (SS) feature similar to that of $Cu(OH)_2$, supporting the view that small Cu clusters of a few Cu atoms (e.g., 2–4 atoms) or widely spaced sheets similar to $Cu(OH)_2$ exist in the material. Subsequently, $Cu(OH)_2$ was employed as a model to fit the EXAFS spectrum of Cu-NU-1000, evidencing that within the MOF Cu is coordinated with four first-shell O atoms with an average Cu–O distance of ~1.94 Å and a square planar configuration, while two further O atoms are located at >2.3 Å from the copper sites. In addition, the fit indicated Cu-Cu scattering at about 2.93 Å with an average CN equal to 1.3 ± 0.3, suggesting that copper may be present in Cu-NU-1000 as Cu-hydroxo dimers, trimers, or tetramers [71]. To further support these

hypotheses, the authors used DFT to optimize the geometry of a $[Cu_3(OH)_4]^{2+}$ fragment inserted and anchored by two hydroxyl groups of each MOF node, and found reasonable agreement between the associated theoretical and experimental EXAFS spectra.

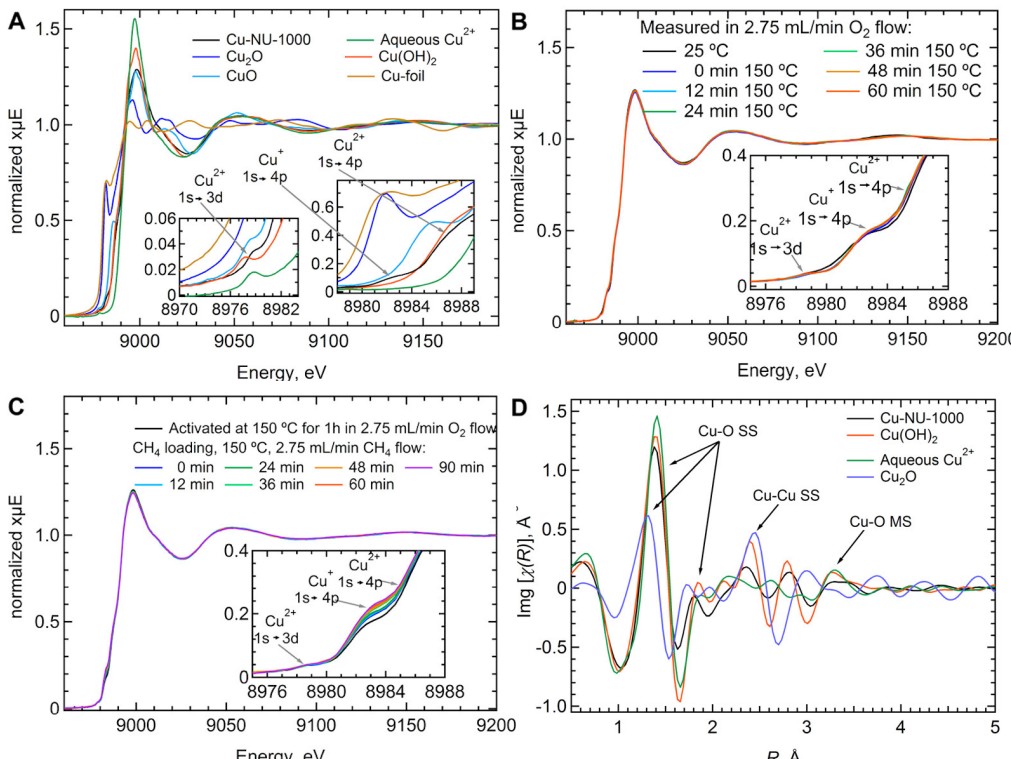

**Figure 14.** (**A**–**C**) Normalized Cu-XANES spectra (**A**) and k²-weighted Cu-EXAFS Img[χ(R)] spectra (**C**) of ALD-synthesized Cu-NU-1000 and reference materials. In panel (**C**), arrows indicate the spectral regions most affected by Cu-O single scattering (SS) and multiple scattering (MS), as well as by Cu-Cu SS. (**B**–**D**) Normalized Cu-XANES spectra of Cu-NU-1000 during activation in oxygen (**B**) and methane loading (**D**) at 150 °C. In panels (**A**,**B**,**D**) the insets evidence the pre-edge feature at 8977–8978 eV due to the 1s→3d electronic transition for the distorted symmetry of Cu(II) as well as the 1s→4p transitions for Cu(I) and Cu(II) at 8982–8984 and 8985.5 eV, respectively (adapted with permission from Ref. [71]. Copyright 2017, American Chemical Society).

Baek and coworkers [69] utilized N K-edge XANES, Cu K-edge XAS, DR UV–Vis, and resonance Raman spectroscopy measurements to obtain detailed structural and electronic information on the investigated three MOF–808–L–Cu catalysts, with the MOF–808–Bzz–Cu species being the most active towards MTM conversion (see Figure 15A). The anchoring of copper sites to N atoms that are part of the imidazole linkers is evidenced by N K-edge XANES spectra. In fact, as one may observe in Figure 15C, two absorption bands at 398.8 and 400.6 eV assignable to 1s→π* transitions are present in the spectrum of MOF–808–Bzz, whereas upon metalation the peaks are shifted and their intensities modified, suggesting that the copper atoms are indeed coordinated by the N-linker atoms. Since the ex situ N K-edge XANES spectra measured after each reaction step remain similar (Figure 15C), it could be concluded that Cu atoms are coordinated with N-linker atoms during the entire MTM process.

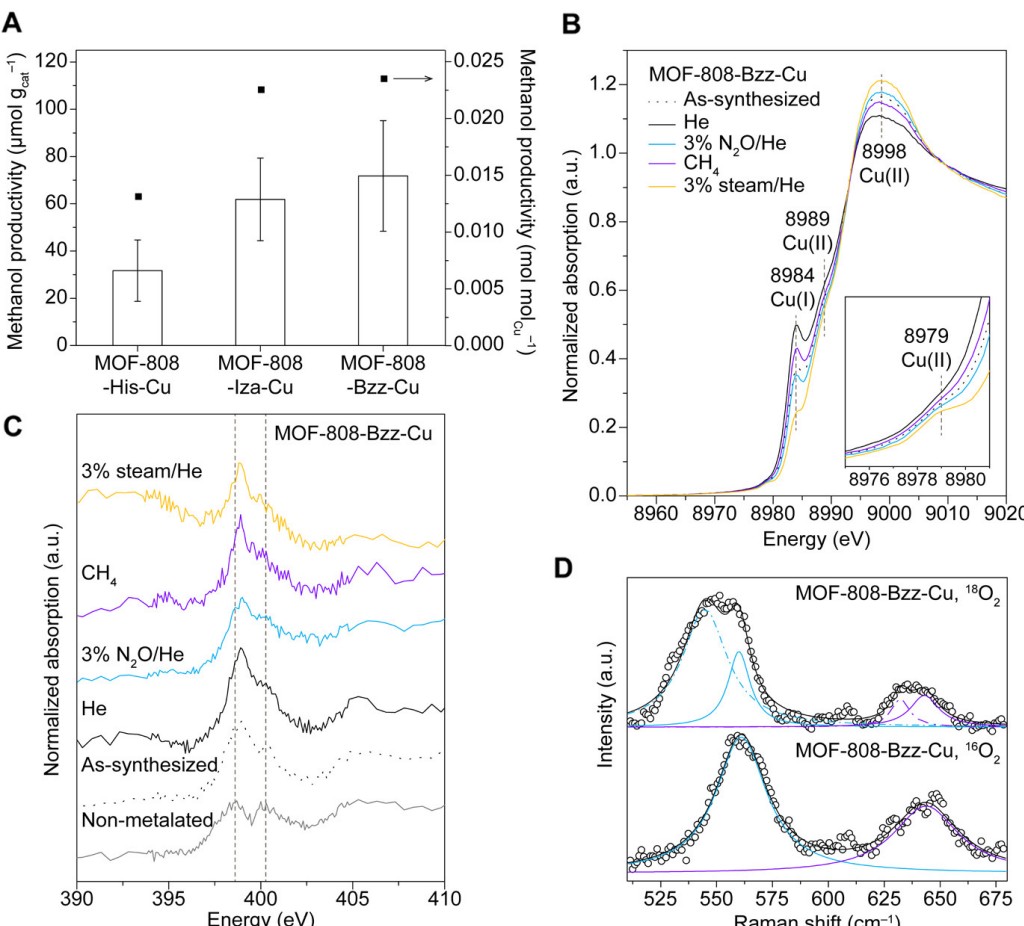

**Figure 15.** (**A**) Average CH$_3$OH productivity of MOF-808-His-Cu, MOF-808-Iza-Cu and MOF-808-Bzz-Cu. (**B**) Ex situ Cu K-edge XANES spectra of MOF-808-Bzz-Cu collected after performing the reactions with He, 3% N$_2$O/He, CH$_4$, and 3% steam/He. (**C**) Ex situ N K-edge XANES spectra of MOF-808-Bzz, as-synthesized MOF-808-Bzz-Cu, and MOF-808-Bzz-Cu after performing the reactions with He, 3% N$_2$O/He, CH$_4$, and 3% steam/He. (**D**) Resonance Raman spectra of MOF-808-Bzz-Cu synthesized using $^{16}$O$_2$ and $^{18}$O$_2$ (the wavelength of the laser was 407 nm) (reprinted with permission from Ref. [69]. Copyright 2018, American Chemical Society).

The ex situ Cu K-edge XANES spectrum of the as-synthesized MOF–808–Bzz–Cu sample exhibited a weak 1s→3d adsorption peak at 8979 eV and a 1s→4p feature at 8984 eV attributable to Cu(II) and Cu(I) species, respectively, a mixture of which is present in the catalyst (Figure 15B). Pre-treating the MOF in He at 150 °C and subsequently exposing the material to 3% N$_2$O/He at the same temperature resulted in XANES modifications indicative of an initial Cu(II) to Cu(I) reduction, followed by the formation of active copper–oxygen species, respectively. After exposing the active MOF to CH$_4$, the Cu(I) peak at 8984 eV increased with a contextual decrease in the white line intensity, suggesting that Cu(II) is reduced to Cu(I) during the reaction. Furthermore, flowing 3% steam/He at 150 °C onto the catalyst in order to desorb methanol resulted in an intensity increase of the Cu(I) peak at 8984 eV, as shown in Figure 15C. Resonance Raman spectroscopy was used to identify the active copper species arising upon exposing the MOF to 3% N$_2$O/He. It was found that the Raman spectra of MOF–808–L–Cu samples oxygenated with either $^{16}$O$_2$ or $^{18}$O$_2$ display isotope-dependent Raman peaks at ∼560 cm$^{-1}$ and ∼640 cm$^{-1}$, and at 545 cm$^{-1}$ and 630 cm$^{-1}$, respectively, which are assigned to Cu–O bond vibrations in the core breathing mode of bis($\mu$-oxo) dicopper species (see Figure 15D). The presence of such dicopper species in the frameworks of all three MOF–808–L–Cu structures was also supported by DFT calculations and EXAFS analyses.

Zheng and coworkers [70] employed in situ XAS, FTIR, and EPR to probe the local variations of the active copper centers along the MTM reaction cycle in cation exchange-synthesized Cu-NU-1000. In particular, the authors investigated MOF catalysts with a Cu percentage in the 0.6–2.9 wt % range. XAS suggested that in the as-synthesized MOFs Cu exhibits a local structure similar to that of Cu in fully hydrated $Cu(OH)_2$. After activating the MOF samples in $O_2$ at 200 °C, XANES and EXAFS spectra evidenced the removal of $H_2O$ ligands coordinating the copper sites, retaining the overall structure, nuclearity, and oxidation state of copper in the pristine materials. In particular, within the MOF exhibiting the highest Cu loading, mainly dinuclear copper species were found, identified predominantly as Cu oxyl-like and cupric hydroxide-like species, while the MOF with the lowest Cu loading predominantly contained mononuclear Cu sites. DFT predictions, in agreement with the collected spectroscopic data, suggested that the dinuclear Cu species have one Cu atom anchored to two $\mu_3$-OH groups of the $Zr_6$ node and a second more distant Cu atom bridged to the first Cu center by means of two $\mu$-OH groups [70].

Ren and coworkers [108] employed XPS to track the evolution of the Cu sites in $CuCl_2$@UiO-bpy and $Cu_xO_y$@UiO-bpy before and after $CH_4$ loading. From the analysis of the Cu 2p and Cu LMM XPS spectra, it was found that in the as-synthesized $CuCl_2$@UiO-bpy framework there is a relative content of Cu(II) and Cu(I) equal to 84 and 16%, respectively. After activating the MOF in $O_2$, the Cu(II) increased to 94%, while after $CH_4$ loading a partial reduction of Cu(II) was observed, with an increase in the relative Cu(I) content from 6% up to 11%. The nature of the $Cu_xO_y$ clusters located inside the MOF framework was assigned on the basis of DFT calculations, which suggested that the $Cu_xO_y$ clusters consist of 0.34% $Cu_3O_3$, 80.85% $Cu_4O_4$, and 18.81% $Cu_5O_5$ in UiO-bpy after $O_2$ activation, with the MTM conversion being more energetically favorable than the $Cu_4O_4$ species.

Lastly, Lee and coworkers [110] leveraged XAS and XPS to characterize the Cu/ZIF-7 catalysts active toward the MTM reaction. The oxidation state of the Cu sites in the as-synthesized MOF was determined to be +2 from the energy positions of the main XAS absorption edge and the pre-edge XAS transition, as well as from XPS data. Wavelet-transform Cu-EXAFS analyses indicated the absence of Cu–Cu bonds, suggesting that mononuclear Cu sites are dispersed within the zeolitic framework, while Cu-EXAFS fitting indicated a $CuN_4$ first-shell coordination with identical Cu–N bond lengths equal to $\sim$1.97 Å.

## 6. Conclusions and Perspectives

In summary, we provided a brief overview of the state of the art concerning the application of MOFs with iron and copper active sites to accomplish the oxidation of methane to methanol, with a particular focus on the spectroscopic techniques employed by authors to unravel the electronic and structural properties of the MOF catalysts. Despite the field being relatively young, several successful studies have reported accomplishing the MTM reaction over iron- and copper-loaded MOFs in both batch and flow reactors. The amalgamation of varied spectroscopies, substantiated by theory, has proven essential to understanding the mechanistic details of reactions at a microscopic level. Nonetheless, it appears clear that the development of cost-efficient MOF catalysts for the MTM conversion still requires significant advances. For instance, the more widespread use of mild experimental conditions and environmentally benign oxidants (such as $O_2$ or $H_2O_2$, which only produces $H_2O$ as a byproduct) in the near future may allow one to access MTM conversion processes over MOFs that are more economically feasible. Furthermore, the very high degree of tunability of MOF constituents may be further leveraged to design active site pockets of variable pore aperture sizes, in an effort to better control MTM selectivity [124], to incorporate carefully chosen linker functionalities that may act as reaction co-catalysts, such as N-heteroaromatic carboxylic acids [6,125], or to increase the hydrophobicity of the MOF inner cavities in order to favor the release of the hydrophilic methanol product. On the experimental side, synchrotron-based advanced techniques, such as, among others, resonant and non-resonant X-ray emission spectroscopy (XES), may be employed, in addition to XAS or the laboratory

spectroscopies explored to date, in order to further complement the MTM reaction picture. The theoretical treatment of MOF catalytic sites for the MTM reaction, which is mainly based on DFT methodologies, may also significantly benefit from the application of multi-reference (MR) approaches. Furthermore, with the increasing use of data-driven theoretical models by the scientific community, we expect that machine learning (ML) and artificial intelligence will provide significant contributions to the efficient in silico screening and selection of promising, new MOF frameworks with desirable properties, to accomplish the MTM conversion. In this regard, the availability of diverse spectroscopic datasets related to the given MOF catalyst's key intermediates will be critical to training ML models.

In conclusion, MOFs with bioinspired iron and copper active sites provide unique opportunities to take on the challenge of efficiently converting methane to methanol, while the combination of advanced and complementary spectroscopic techniques allows access to the reaction mechanistic details, which are required to rationally improve MOF catalysts for the MTM conversion.

**Author Contributions:** Conceptualization, F.T., A.T. and P.D.; writing—original draft preparation, F.T. and A.T.; writing—review and editing, F.T., A.T. and P.D.; supervision, P.D. All authors have read and agreed to the published version of the manuscript.

**Funding:** This research received no external funding.

**Acknowledgments:** The authors acknowledge the European Union-NextGenerationEU under the Italian Ministry of University and Research (MUR), Network 4 Energy Sustainable Transition—NEST project (MIUR project code PE000021, Concession Degree no. 1561 of 11 October 2022)—CUP C93C22005230007.

**Conflicts of Interest:** The authors declare no conflict of interest.

## Abbreviations

The following abbreviations are used in this manuscript:

| | |
|---|---|
| GWP | global warming potential |
| MTM | methane to methanol |
| BDE | bond dissociation energy |
| MOF | metal–organic framework |
| BDC | benzene-1,4-dicarboxylate |
| DOBDC | 2,5-2,5-dioxido-1,4-benzenedicarboxylate |
| BTC | benzene-1,3,5-tricarboxylate |
| ABTC | 3,3′,5,5′-azobenzene-tetracarboxylate |
| TBAPy | 1,3,6,8-tetrakis(p-benzoate)-pyrene |
| SBUs | secondary building units |
| DFT | density functional theory |
| sMMO | soluble methane monooxygenase |
| NADH | nicotinamide adenine dinucleotide |
| HTS | hydrothermal synthesis sample |
| ECS | electrochemical synthesis sample |
| LL | low loading |
| HL | high loading |
| TOF | turnover frequency |
| TON | turnover number |
| bpy | 2,2′-bipyridine |
| $H_2$bpydc | 2,2′-bipyridine-5,5′-dicarboxylic acid |
| XPS | X-ray photoelectron spectroscopy |
| EPR | electron paramagnetic resonance |
| XAS | X-ray absorption spectroscopy |
| pMMO | particulate methane monooxygenase |
| QM/MM | quantum mechanics/molecular mechanics |
| EDS | energy-dispersive spectroscopy |
| XANES | X-ray absorption near edge structure |

| DRS | diffuse reflectance spectroscopy |
| PXRD | powder X-ray diffraction |
| HAADF-STEM | high-angle annular dark-field scanning transmission electron microscopy |
| PDF | pair distribution function |
| EXAFS | extended X-ray absorption fine structure |
| FT | Fourier transform |
| CN | coordination number |
| SS | single scattering |
| MS | multiple scattering |
| XES | X-ray emission spectroscopy |
| MR | multi-reference |
| ML | machine learning |

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
