# Peer review of "Exploring the Methane to Methanol Oxidation over Iron and Copper Sites in Metal–Organic Frameworks"

_catalysts, doi:10.3390/catal13101338_

Round 1

Reviewer 1 Report

1. Captions of Table 2 should be on top of table.

2. The information in the manuscript is not clear and organized well. For example, in table 3 when the authors list various spectroscopic methods, it’s better to include relevant references for each method. Besides, the authors should include significant findings or key results for each characterization with corresponding references. This kind of table organization applies to other sections such as Fe and Cu active site-containing MOFs.

3. The authors claims to explore the synthetic strategies employed to incorporate iron and copper sites into different MOF. Can they point out?

4. The authors mentioned bio-inspired MOF, but there’s a lack of connection statements between MOF and bio-inspired one. Can they provide the information?

5. Overall, the manuscript requires a more coherent and concise organization.

Ok 

Author Response

See file attached

Reviewer 2 Report

The manuscript is well prepared and interesting which is helpful for developing efficient catalysts in MTM. The reviewer suggests that the preparation methods of the two kinds of iron and copper involved catalysts can be included in the two sections, respectively.

Specific comments:

-The authors review  the application of MOFs with iron and copper  in catalytic MTM reaction. Due to the significance of the reaction which converts methane to methanol, this review will be focused on.
-The MOF support provides space to accomodate the active sites and protects them against leaching.
-
Recently as a novel kind of support, the relavant reseaches on MOF-supported catalysts will open a path to design the efficient catalyst.
-The addition of the preparation methods of such catalysts in the manuscript has been suggested.
-The conclusions and the perspectives are consistent and plausible.
-Although there are many figures and tables, in my opinion, these selections are requireable.

Author Response

See file attached
